# In Males with Adequate Dietary Needs Who Present No Sleep Disturbances, Is an Acute Intake of Zinc Magnesium Aspartate, Following Either Two Consecutive Nights of 8 or 4 h of Sleep Deprivation, Beneficial for Sleep and Morning Stroop Interference Performance?

**DOI:** 10.3390/bs14070622

**Published:** 2024-07-22

**Authors:** Ben J. Edwards, Ryan L. Adam, Chloe Gallagher, Mark Germaine, Andrew T. Hulton, Samuel A. Pullinger, Neil J. Chester

**Affiliations:** 1Research Institute for Sport and Exercise Sciences, Liverpool John Moores University, Liverpool L3 3AF, UKc.gallagher@2022.ljmu.ac.uk (C.G.); n.chester@ljmu.ac.uk (N.J.C.); 2School of Health and Human Performance, Dublin City University, D09 V209 Dublin, Ireland; mark.germaine2@mail.dcu.ie; 3Department of Nutrition, Food, and Exercise Science, University of Surrey, Surrey GU2 7XH, UK; a.hulton@surrey.ac.uk; 4Sport Science Department, Inspire Institute of Sport, Vidyanagar, Bellary 583275, India; samuel.pullinger@inspireinstituteofsport.com

**Keywords:** supplementation, cognitive function, micronutrients, individualised response, sleep restriction

## Abstract

Purpose: Purpose: We examined whether supplementation of zinc magnesium aspartate (ZMA) in two groups of males, either partially sleep-restricted (4 h) or with habitual sleep (8 h) for 2 nights, was beneficial for sleep and subsequent morning Stroop performance. Methods: Participants were randomly allocated to two independent groups who either had 4 h (33 males) or 8 h (36 males) sleep for two nights. Using a double-blinded, randomised counterbalanced design, they then completed five sessions, (i) two familiarisation sessions including 7 days of sleep and dietary intake, (ii) three conditions with 4 h or 8 h sleep and either NoPill control (NoPill), placebo (PLAC) or ZMA (ZMA). Sleep was assessed by actimetry and sleep questionnaires, and cognitive performance was assessed by the Stroop test. The data were analysed using a general linear model with repeated measures. Results: A main effect for “sleep” (4 or 8 h) was found, where more opportunity to sleep resulted in better “sleep” metrics (both objective and subjective) as well as better Stroop scores (lower colour-interference and word-interference scores and lower error in words). No main effect for “Pill” was found other than the mood state depression, where subjective ratings for the PLAC group were lower than the other two conditions. Interactions were found in anger, ease to sleep and waking time. Conclusion: Having 8 h opportunity to sleep resulted in better “sleep” metrics as well as better Stroop scores compared to 4 h. Supplementation of ZMA for 4 or 8 h for 2 nights had no effect on subsequent morning cognitive performance but reduced sleep or total sleep time by ~0.46 h compared to the other conditions. An interaction was found where sleep time was reduced by ~0.94 h in the ZMA group in the 8 h condition compared to NoPill or PLAC.

## 1. Introduction

Sleep plays a major role as a recovery strategy in athletes, where less than the recommended amount (such as 7–9 h for 18–60 year olds [1]) leads to detrimental alterations in cellular restitution, growth and repair, brain detoxification, consolidation of memories and immune function [2,3,4,5]. Partial sleep loss or sleep deprivation is classed as a reduction in sleep per night, compared to that habitually taken in a 24 h period. According to Banks and Dinges [6], partial sleep deprivation can occur in three ways: (1) preventing the normal progression and sequencing of sleep stages from being physiologically consolidated relative to time in bed, called sleep fragmentation; (2) loss of specific physiological sleep stages, referred to as selective sleep stage deprivation, which can occur if sleep fragmentation is isolated to a specific sleep stage (such as when apnoeic episodes disrupt primarily one stage of sleep such as REM sleep, or when medications suppress a specific sleep stage); and (3) sleep restriction, which is also referred to as sleep debt, which is characterised by reduced sleep duration. Reduced sleep over several days (hence sleep disturbance by restriction) is a common occurrence in athletic and non-athletic populations, resulting in reduction in sleep quality and quantity [7,8,9]. Athletes might experience sleep loss for many reasons, such as during daily training where early rising or retiring late at night is required, apprehension the night before competition and sleeping in unfamiliar surroundings. Combined with time zone transition disturbance and environmental and psychological factors, athletes are susceptible to achieving <7 h of sleep per night [10]. This can have detrimental effects on mood and motivation, an essential element for tasks requiring higher cognitive function (such as executive functions), especially in the morning, when this is compromised by partial sleep deprivation [8,11]. Executive functions notably include the ability to plan and coordinate considered action regardless of alternatives, to monitor action, update as necessary and to suppress distractions by focusing attention on the relevant information (i.e., inhibition). One of the most widely used neuropsychological tests to study attention and notably its inhibitory processes is the Stroop colour–word test [12,13]. This task has been used extensively to study limitations in the ability to fully suppress the influence of a dominant source of information, such as automatic word reading. Therefore, the magnitude of the Stroop interference has been used as an indicator of the efficiency of the inhibitory function.

Zinc magnesium aspartate (ZMA) is a popular supplement for recreational and elite athletes that has led to several studies investigating the proposed benefits for health and sporting performance [14,15,16], in addition to sleep and circadian regulation. The proposed mechanism by which zinc (Zn) and magnesium (Mg) promote sleep is through the synthesis and function of sleep–wake neurotransmitters such as gamma-aminobutyric acid (GABA), which supports sleep when activated [17]. Evidence of the effectiveness of supplementation of Zn and Mg has predominately focused on populations whose levels are below the recommended daily allowance, either very young or aged and are clinical-based such as insomniacs [18,19]. In athlete populations, the investigation of ZMA has predominantly suffered from a low sample size, lack of prior measurement of diet for macro- and micronutrient quantities, baseline assessment of habitual sleep and resultant sleep after ingestion of the supplement. Furthermore, research design issues are apparent and include the absence of a “Pill” condition, as well as a placebo (PLAC) and experimental (ZMA) condition [20].

Recent work that addressed the concerns about research design has investigated if recreational athletes (n = 16) suffering from partial sleep deprivation (but are otherwise healthy with no sleep disorders and with a balanced diet) may experience improvements in sleep variables with acute ZMA supplementation [20]. It was hypothesised that by supplementing ZMA pre-sleep, reductions in sleep latency and/or fragmentation within the 3–4 h restricted sleep window may result in reduced detriments to morning cognitive performance. However, the authors highlighted no benefits of supplementation on sleep (objectively or subjectively measured) or morning cognition. As well those who are partially sleep deprived, populations with normal/recommended sleep are also supplementing ZMA for its proposed [14] yet uncorroborated effects on subsequent increases in testosterone and force [15,21]. Large-scale research investigating acute ZMA supplementation (1–2 nights) is therefore warranted in populations of both “normal sleepers” with a balanced diets and those who are partially sleep deprived, to further understand the effects on subsequent sleep and morning cognitive performance.

Therefore, we examined in males whether supplementation of ZMA, following 2 nights of either 4 or 8 h of sleep, improved (i) markers of objective and subjective sleep (via actigraphy and sleep questionnaires) and (ii) morning cognitive performance, compared to 2 nights of no pill (NoPill), prescribing a placebo (PLAC) or ZMA capsules. We hypothesised that the group with 8 h of sleep would have better sleep and cognitive performance than those who had 4 h, and ZMA would have no beneficial effects on (a) ‘sleep’ variables and (b) cognitive performance—attention and notably its inhibitory processes ([12,13] Stroop task/test) in our chosen population of healthy male recreational athletes irrespective of the prior sleep taken for 2 nights.

## 2. Materials and Methods

### 2.1. Participants

Sixty-nine males as identified by sex and gender (mean ± SD: 22.8 ± 3.2 years; body mass: 78.4 ± 6.5 kg; body stature: 176.6 ± 6.5 cm; normative retiring and rising times: 23:08 ± 00:34 hh:min and 08:12 ± 00:33 hh:min, respectively) participated in the investigation. These were randomly allocated into one of two groups, which were 4 h of sleep (n = 33) or 8 h of sleep (n = 36, Table 1). Fifteen of the sixty-nine participants’ data (22%) investigating ZMA and 4 h of sleep have been previously published [20]. To assess the first hypothesis that individuals who slept for 8 h per night would have better sleep quality and cognitive performance than those who slept for 4 h, the sample size was determined using power calculation software (G*Power, version 3.1.9.6), assuming an independent t-test (2 groups) based upon a moderate effect size of 0.6 for sleep quality and Stroop interference [12,13] with a power of 0.80 and α = 0.05, which determined that a sample of 36 participants was required. Thirty-six participants were recruited for each group, but three dropped out in the 4 h condition. To test the effects of ZMA on ‘sleep’ variables and cognitive performance using a paired t-test, based upon a large effect size of 0.6 for sleep quality and Stroop number with a power of 0.80 and α = 0.05, a sample of 19 participants was required. Females were omitted from the current investigation to limit biological sex differences due to hormonal variation, as well as females being relatively phase-advanced in their tiredness/alertness rhythms compared to those of males [22], with higher prevalence of difficulty maintaining sleep and early-morning awakenings reported in women vs. men. This could impact on morning mood states and cognitive performance, especially after sleep loss, in a way different from males. As there is little research in this area, we sought to reduce bias. In line with the inclusion criteria, participants were recreationally active (as classified by the “Participant Classification Framework” [23]), injury-free with no diagnosed sleep disorders and had not completed shiftwork or travelled outside the local time zone in the past month. Participants were required to arrive fasted and abstain from alcohol, caffeine and exercise for 24 h preceding a testing session, with no napping between sessions. Exclusion criteria: None of the participants could be receiving any pharmacological treatment throughout the study period. Habitual caffeine consumption was assessed using the caffeine consumption questionnaire (CCQ), and those with <150 mg per day were excluded [24]. Prior to participating in the investigation, participants were presented with an information sheet followed by a ‘Physical Activity Readiness Questionnaire’ (PARQ) [25] and a written consent form. Verbal explanation of the experimental procedure was provided; this included the aims of the study, the possible risks associated with participation and the experimental procedures. Participants were assessed for circadian chronotype using the ‘Composite Morningness/Eveningness Questionnaire’ [26]. The mean chronotype score on a 13–52-point scale was 32.6 ± 3.3; hence, all participants were of the intermediate type. All participants gave their informed consent for inclusion before they participated in the study. Experimental procedures were approved by the University Human Ethics Committee (M21_SPS_1595, date of approval 5 November 2021) and conducted in accordance with the ethical standards of the journal and complied with the Declaration of Helsinki.

### 2.2. Research Design

All participants were required to visit the laboratory on five occasions (dry temperature of 19 °C, 35–45% humidity and a barometric pressure of 750–760 mmHg, respectively). Prior to attending the laboratory, participants completed a 5-day habitual food/fluid diary and weighed food intake, 7-day habitual sleep recording using actigraphy (Motionwatch 8, CamnTech, Co., Dublin, Ireland), in addition to a sleep diary as a secondary measure (Table 1). The values for measurements of daily zinc and magnesium were determined by dietary analysis using the computer program Nutritics (Nutritics V6, Co., Dublin, Ireland [29]); this process was conducted by a SENr-registered Sports and Exercise Nutritionist. The initial laboratory visits involved completion of two visits for familiarisation sessions. Both familiarisations involved collection of participants’ height, mass, completion of questionnaires (Profile of Mood States, POMS [30]; Stanford Sleepiness Scale, [31]; and sleep questions from the Liverpool Jet-lag questionnaire [32]) and completion of the Stroop test (see Figure 1 and the ‘measurements’ section for details). The remaining sessions consisted of three experimental conditions, involving two consecutive nights of prescribed sleeping, sleep restriction (retiring at 02:30 h and rising at 06:30 h) or normal sleep (retiring at 22:30 h and rising at 06:30 h) at the participants’ home, before entering the laboratory at 07:00 h on the third day. Prior to bed the participants either consumed three ZMA or PLAC capsules or NoPill depending on the condition. ZMA capsules contained 30 mg of zinc, 450 mg of magnesium and 10.5 mg of vitamin B6 (PhD Nutrition LTD, Yorkshire, UK), and placebo capsules were made in the department and contained maltodextrin (Sport supplements Ltd. t/a BulkTM, Colchester, UK). Researchers and participants were blinded to the supplement schedule, and pills were provided in a plastic bottle with instructions to consume with water 30 min before retiring. Both ZMA and the placebo were lightly dusted with maltodextrin to create a similar taste; both had similar weight (0.8 g/capsule) and were 00 size. At the end of the experiment, the order of treatment was revealed to the researchers by an author (BE). Before experimental sessions, participants were asked to refrain from vigorous physical activity 24 h prior, during which time they also had to avoid any alcoholic or caffeine-containing drinks. No food was to be consumed 1–2 h before the experimental protocol, for the morning testing session and before sleep. In the two hours before retiring to sleep, participants were asked to refrain from watching television or use of their mobile devices and were also required to consume supplements provided if on the ZMA or PLAC condition. To ensure recovery and to enable wash out for the ZMA between trials, there was at least a week between testing conditions for all participants. The experimental sessions were then counterbalanced in order of administration to minimise any potential learning effects [33], with a minimum of 72 h to ensure recovery between trials. All experiments were completed between the months of October and May (autumn to spring in the UK) with the sunrise and sunset range from the start to the end of the experiment being 05:37 to 07:29 h and 18:01 to 20:40 h, respectively. Testing was supposed to finish in February to ensure the individual’s exposure to sunlight in the mornings prior to entering the laboratories was <80 Lux. Unfortunately, due to COVID-19 restrictions, the time frame had to be extended.

### 2.3. Measurements

Prior to the main experimental laboratory sessions, two familiarisation sessions reduced any learning effects of the Stroop test ([12,13] or word/colour interference test), where the participants were asked to read out their responses to words or colours for 45 s as possible and to leave no errors uncorrected. This was filmed, and the number of errors and total amount completed was recorded and analysed. The first sheet had text (red, blue, yellow, black and green) in black ink (naming word test). The second sheet had blocks of colour corresponding to the text on the first sheet (naming colour test, C). With the third sheet, the participants had to read out the word (which was coloured differently to the word, e.g., the word was yellow and the colour red, referred to as the naming colour of word test, CW), and for the fourth sheet, the participants had to read out the colour (which was wrongly named, e.g., the colour was yellow but the word was red, referred to as the naming of word not colour test, WC). In this fourth sheet, the words were printed in the reverse order to the third sheet. The raw data were analysed for number of errors (to give an indication of pacing/speed accuracy in the 45 s) and interference scores (an indicator of the efficiency of the inhibitory function), where C represents the correct answers produced in 45 s in naming colours, and CW corresponds to the correct answers achieved in 45 s in the interference condition [34]. This was also performed for the naming of word not colour test.
I = [((C − CW)/(C + CW)) × 100](1)

Following two consecutive nights, participants arrived at the laboratory at 07:00 h and sat having risen at 06:30 h; after 30 min, they completed the rating of mood questionnaire (POMS), sleep questionnaires and sleepiness questionnaire. Participants then undertook the Stroop test (Figure 1). To monitor sleep across the two consecutive nights, participants put an actiwatch on their nondominant wrist in the evening, prior to the ingestion of the “pill” (Motionwatch 8, CamnTech), and data were downloaded for analyses on the 3rd morning when participants arrived at the laboratory for testing. In between experimental conditions, participants were under ‘normal living’ conditions.

### 2.4. Statistical Analysis

The Statistical Package for the Social Sciences (SPSS IBM) version 28 for Windows was used. Differences between conditions were evaluated using a general linear model with repeated measures, between the subject factor for “Sleep opportunity” (4 or 8 h sleep; 2 levels), within the subject factor for “Pill” (NoPill, PLAC, ZMA; 3 levels) and interaction. To correct violations of sphericity, the degrees of freedom were corrected in a normal way, using Huynh–Feldt (ε > 0.75) or Greenhouse–Geisser (ε < 0.75) values for ε, as appropriate. Graphical comparisons between means and Bonferroni pairwise comparisons were made where main effects were present. The α level of statistical significance was set at *p* < 0.05. Effect sizes (Cohen’s d) were calculated from the ratio of the mean difference to the pooled standard deviation. The magnitude of “d” was classified as trivial (≤0.2), small (>0.2–0.6), moderate (>0.6–1.2), large (>1.2–2.0) and very large (>2.0) based on guidelines from Batterham and Hopkins [35]. Pearson correlations between sleep difference (h) and Mg or Zn levels (either mg or mg/kg body mass) for ZMA-NoPill for the 8 h sleep condition were undertaken. The results are presented as the mean ± standard deviation (SD) throughout the text unless otherwise stated. Ninety-five-percent confidence intervals (CIs) are presented where appropriate as well as the mean difference between pairwise comparisons.

## 3. Results

### 3.1. Evening Physiological and Psychological Variables

Mean ± SD values and the results from the ANOVA statistical analysis are displayed in Table 2 and Table 3. Statistical significance of the results can be seen in Figure 2.

### 3.2. Measures of Sleep

#### 3.2.1. Actigraphy Variables

There was a significant group effect for opportunity to sleep (“sleep time”) where sleep onset latency (mean difference of 5.4 decimal min; 95% CI: 1.3–9.4, *p* < 0.010, d = 0.55) and actual sleep time (2.74 decimal h; 95% CI: 2.5–3.0, *p* < 0.001, d = 1.79) were higher at 8 h than 4 h opportunity (Table 2). There was a significant main effect of “Pill” for actual sleep time where the ZMA condition led to a reduced sleep time compared to NoPill (0.44 decimal h; 95% CI: 0.2–0.7, *p* < 0.001, d = 0.26) and a reduced sleep time compared to PLAC (0.49 decimal h; 95% CI: 0.3–0.7, *p* < 0.001, d = 0.30). There was a significant interaction of “Sleep” and “Pill” for actual sleep time where profiles of 4 and 8 h for NoPill and PLAC are similar, but during the 8 h condition, opportunity to sleep is reduced in the ZMA condition compared to NoPill and PLAC (~0.94 h, d = 3.08, Table 2).

#### 3.2.2. Waterhouse and Stanford Sleepiness Sleep Questionnaires

There was a significant effect for “sleep time” for ease to sleep, time to sleep, waking time and alertness, where the last night’s sleep compared to normal was easier to get to sleep in the 4 vs. 8 h condition (1.2 AU, *p* < 0.001, d = 0.46), at a later time (2.5 AU, *p* < 0.001, d = 0.88), with more waking episodes in the 8 h condition (−1.1 AU, *p* < 0.001, d = 0.49), at an earlier waking time in the 4 h condition than the 8 h (2.1 AU, *p* < 0.001, d = 1.02) and with greater alertness 30 min after waking in the 8 h condition (1.7 AU, *p* < 0.001, d = 0.82; Table 2). Degree of sleepiness was higher in the 4 vs. 8 h condition (1.34 AU, *p* < 0.001, d = 1.04; Table 2). There was no significant main effect of “Pill” on subjective sleep or sleepiness ratings (*p* > 0.05). There was a significant interaction of “Sleep” and “Pill” for waking time where profiles of 4 and 8 h for NoPill and PLAC are similar, but in the ZMA condition, during the 8 h opportunity to sleep, waking time is considered as later than normal, and during the 4 h opportunity, earlier (0.85 AU, d = 0.40, Table 2). Further, ease of sleeping profiles of 4 and 8 h for NoPill and PLAC are similar, but in the ZMA condition, during the 8 h opportunity, ease of sleeping was harder, and in the 4 h opportunity, it was easier than normal to get to sleep (2.29 AU, d = 0.82, Table 2).

### 3.3. Profile of Mood State

There was a significant effect for “sleep time”, where Vigour and Happy were lower and Tension, Confusion, Depressed and Fatigued values were higher in the 4 h than the 8 h sleep opportunity (*p* < 0.05, d = 0.45–0.9, Table 3). A significant main effect of “pill” was present for the mood state Depressed, where values were generally lower in the PLAC condition than the NoPill (0.53 AU, 95% CI: 0.05–1.01, *p* = 0.024, d = 0.06) or ZMA (0.44 AU, 95% CI: 0.02–0.85, *p* = 0.035, d = 0.897). Only Anger showed an interaction for “sleep time” by “Pill”, where the profiles are higher in the 4 h condition than 8 h for NoPill and PLAC but are at a similar level for ZMA irrespective of 4 or 8 h opportunity to sleep (*p* > 0.05; d = 0.058, Table 3).

### 3.4. Stroop (Colour–Word, Word–Colour Interference Test)

There was a significant effect for “sleep time”, where there was a lower interference score for colour (2.0; 95% CI: 1.0–3.0, *p* < 0.001, d = 0.85), and word (1.7; 95% CI: 0.694–2.7, *p* < 0.001, d = 0.61) and lower word number of incorrect responses/errors in the 8 h group (1.3 AU, *p* = 0.034, d = 0.57). There was no significant main effect of “pill”, nor interaction for “sleep time” by “Pill” (*p* > 0.05; Table 3 and Figure 2).

### 3.5. Pearson Correlations between Sleep Difference (h) and Mg or Zn Levels (Either mg or mg/kg Body Mass) for ZMA-NoPill for 8 h Sleep Condition

A small negative correlation (r = −0.203, *p* = 0.236) which was not significant was found for Mg, and a trivial negative correlation (r = −0.106, *p* = 0.538), again not significant was found for Zn. The strength of the relationship was not affected when controlling for body mass for Mg (r = −0.213, *p* = 0.220) or Zn (r = −0.178, *p* = 0.306).

## 4. Discussion

The results of the investigation found that the group with the 8 h compared to the 4 h opportunity to sleep reported higher positive levels of subjective mood states for Vigour and Happy as well as lower Tension, Confusion, Depression and Fatigue values (d = 0.45–0.9, Table 3). Zn and Mg supplementation, within non-athletic populations, has also been previously shown to improve mood states, yet unlike caffeine, the mechanisms and understanding are still to be fully understood [36,37,38,39]. Only the mood state “depressed” showed a main effect of “pill” where PLAC showed lower values than both NoPill and ZMA; the effect size for this was small (d = 0.43). Actual sleep taken was 6.22 vs. 3.48 h between the two conditions (d = 1.79), with sleep onset latency being shorter in the 4 h than the 8 h condition. For the 8 h compared to the 4 h condition, the subjective questionnaire items reported greater ease of sleeping, quality of sleep and time to sleep and a lower degree of sleepiness and ratings of alertness 30 min after waking. Further, morning Stroop interference scores are an indicator of the efficiency of the inhibitory function and were better for the 8 h condition, with colour–word and word–colour values being lower and word error lower.

These findings were expected in agreement with the previous literature investigating the effect of partial sleep deprivation by restriction on sleep, as well as some measures of verbally mediated processing speed and executive functioning [8,40]. Sleep architecture has been shown to be altered by sleep restriction, but all sleep stages are not affected equally. Similar to the current findings of sleep onset latency being shorter in the 4 h than the 8 h condition reflecting the higher accrued sleep debt and homeostatic drive to sleep on the second night (~5.39 decimal mins, d = 0.51), Banks and Dinges [6] reported healthy adults fell asleep quicker and had decreased time in NREM stage 2 sleep and REM sleep when restricted to 4 h of nocturnal sleep for multiple nights. However, no decrease in NREM slow-wave sleep (SWS) relative to a typical 8 h nocturnal sleep period was identified. Similarly, the current investigation that employed a protocol of 50% sleep loss for two nights found no change in sleep efficiency and fragmentation index between the 4 and 8 h sleep groups (Table 2). The “lapse hypothesis” has been the predominant explanation for sleep-loss-related performance, where decrements in performance due to sleep deprivation are attributed to brief periods of unresponsiveness that increase in frequency as a function of hours of sleep loss. This theory states that lapses are caused by lowered arousal levels, where arousal decreases gradually with sleep loss, and if it falls below a certain level, microsleeps occur (brief bouts of sleep that intrude into wakefulness) [41]. However, as this model cannot fully explain a reduction in cognitive function with sleep loss, a revised model “state instability” hypothesis has been proposed, where variability in performance changes reflecting the interaction of the homeostatic drive for sleep and the endogenous circadian drive for wakefulness, as well as the individual effort of the participant to perform [42]. Although the prefrontal cortex may be susceptible to the effects of sleep loss, research investigating sleep deprivation on executive functional tasks (measures of prefrontal functioning) show inconsistent findings. Sleep loss may affect specific cognitive systems above and beyond the effects produced by cognitive declines or impaired attentional processes [43]. The two attentional systems involved in the Stroop task are (i) the anterior attentional system associated with the dorsolateral prefrontal cortex, which assists naming/task relevant information and other executive functions, and (ii) the posterior attentional system (anterior cingulate cortex), which selects appropriate responses and allocates attentional resources. These systems may be differentially affected by sleep loss or capable of engaging in compensatory recruitment. In agreement with the current findings, the Stroop task has been shown to be sensitive to 100% sleep loss (after a total of 24 and 36 h total sleep deprivation) in male or young healthy male and female participants [44,45]. Further, in the current investigation, we found morning colour–word Stroop interference scores (indicator of the efficiency of the inhibitory function) were better for the 8 h than the 4 h condition. But we also showed the reverse Stroop (word–colour) interference values to be sensitive to sleep loss, where the participants read the word that was printed in a different colour to the word. These levels of interference were lower than the colour word test as expected, as the basis of the “Stroop effect” relies on the fact that humans have trouble naming a physical colour when it is used to spell the name of a different colour—such that there is a delay in reaction time between congruent and incongruent stimuli. Lastly, we found that with the 8 h sleep, not only were word–colour interference values lower but so was the error, hence an associated accuracy and speed reduction with sleep loss. On the other hand, others have found no effect of short-term sleep deprivation on the Stroop task after 34–36 h total sleep deprivation, suggesting that sleep deprivation does not selectively impair prefrontal functioning, notably the cognitive flexibility and the capacity to shift from one response set to another [46,47]. However, in the supporting literature, confounding factors such as fatigue, stress and different amounts of sleep debt as well as the method used to analyse interference or type of Stroop (colour–word, emotional and specific) as well as using a fixed time (or not) might have affected performance.

In the current investigation, a pragmatic approach was adopted involving a protocol that assessed the effects of an acute sleep disruption (commonly found in athletes), often associated with travel to competition or training [5,8]. This procedure has been utilised by others [9,10,20,48]. Acute ZMA supplementation for two nights in recreationally active males, who through habitual diet met the RDA for the micronutrients in question, did not improve sleep (as measured by actigraphy and sleep questionnaires). It was hypothesised that ZMA improves sleep by a decrease in sleep latency and an increase in the quantity of slow-wave sleep [49,50]. Further, magnesium may act to increase the activation of GABA neurotransmission, which contributes to improvement in sleep architecture, particularly slow-wave sleep, which is associated with restorative sleep [51,52]. Rather, in our ZMA condition, the total sleep time was reduced by ~0.46 h compared to the other conditions, where the magnitude of this effect size was small (d = 0.28). However, an interaction was found where sleep time was reduced by ~0.94 h in the 8 h condition with a very large effect size (d = 3.08). Although polysomnography was not employed to establish sleep directly, these findings suggest that in a population of good sleepers (~8 h a day with no sleep disorders) with a balanced diet, ingestion of ZMA causes a reduction in actual sleep during the 4 h opportunity but especially during the 8 h. There was no modification of sleep efficiency or fragmentation index, as these variables did not change in the investigation with the reduced sleep time. This finding is worth further consideration as it concerns most of the athletic population who use ZMA to aid in sleep, who are sleep deprived.

Oral uptake of Zn and Mg is essential for survival, with both nutrients previously highlighted to improve sleep parameters in healthy and unhealthy elderly populations who have poor dietary intake, such that they have micronutrient deficiencies [53,54]. The recommended dietary allowance (RDA) for zinc is 9.5 mg/day for men (aged 19 to 64 years [27]) and for Mg is 420 mg/day for men [27]. In the investigation population, micronutrient intake (food and supplement) was similar for the 4 and 8 h groups (Zn intake of 22.4 vs. 23.2 mg/day, Mg of 716.6 vs. 750.2 mg/day and B6 [pyridoxine] of 4.4 vs. 4.3 mg/day). No significant correlations were found for Mg or Zn (r = −0.203, *p* = 0.236; r = −0.106, *p* = 0.538) and change in total sleep between ZMA and NoPill values for the 8 h condition (which showed greater sleep loss with ZMA ~0.94 h; Figure 3). Zn and Mg levels ingested through diet and supplementation were ~31.9 mg or 336% RDA for Zn and ~1016.6 mg or 339% RDA for Mg. The strength of the relationship was not affected when controlling for body mass for Mg (r = −0.213, *p* = 0.220) or Zn (r = −0.178, *p* = 0.306), where 1.3 and 1.1% of the variation in Zn or Mg intake can account for the change in actual sleep time (Figure 3). In future research, greater spread of levels of Zn and Mg supplementation including values below and above the required RDA would give a better understanding of the relationship of these essential minerals and resultant sleep. What has received less attention is the toxicology of supplementation and dose, the pharmacological effect on sleep and cognitive performance, which, in the current investigation, was ~336% for Zn and ~339% for Mg compared to the RDA for men. Consuming high doses of Zn reduces the amount of copper the body can absorb. This can lead to anaemia and weakening of the bones, whilst high doses of magnesium (more than 400 mg) for a short duration can cause diarrhoea. Participants were not asked to report any side effects directly related to the supplements, it was established that volunteers were not taking any other dietary supplements nor presenting flu like symptoms, as over-the-counter Zn lozenges and nasal sprays are commonly used for the treatment of the common cold and can add to the total Zn intake. Research conducted on the use of ZMA regarding the effects on sleep is scarce [14,15,16]. In a similar study investigating the use of ZMA to aid sleep following sleep restriction in individuals without pre-existing sleep disorders and/or nutrient deficiencies, Gallagher et al. [20] reported no benefit of taking ZMA for 2 nights compared to NoPill or PLAC on sleep parameters, but reported for the Stroop test that the NoPill condition achieved the lowest total score (number of answers) for response of words with no change in the number of errors when compared to ZMA. The highest total score was achieved in the PLAC condition, demonstrating that ZMA did not influence cognitive ability. This finding supports the use of a condition where there was no physical “Pill” ensuring that any placebo effect is accounted for and that the true potential effect of ZMA can be established. In the current study, we incorporated the participants from Gallagher [20] into the current larger-sized study. However, rather than total scores, we calculated interference for CW and WC and errors were analysed from the Stroop test. No difference was found for pill or no pill, or interaction for condition of sleep (4 or 8 h) or pill.

Finally, one of the few studies investigating the relationship of Zn and cognitive performance in older adults (55–87) found that varying doses of Zn (15 or 30 mg/day) used for 3 months had a beneficial effect on spatial working memory [55]. Currently, the scientific literature indicates several cognitive domains such as attention (division and switching of attention), perception and processing of information (fast reaction time) and visual–spatial skills (navigation in a virtual environment) [56]. A full analysis of cognitive performance not only for attention as we have conducted and effects of ZMA in restricted and habitual sleep is warranted in future research.

Further comparison between the current investigation and others is limited. Only one investigation to the authors’ knowledge has utilised a study design that incorporated a NoPill group in conjunction with the ZMA and PLA group, ensuring that any placebo effect is accounted for and that the true potential effect of the supplement can be established [20]. There is limited research comparing the quantities of Mg, Zn and B6 (450 mg of Mg, 30 mg of Zn and ~11 mg of B6) and the recruitment of recreational athletes [15,20]. Further, in only two studies was the habitual dietary intake of participants reported, with data for the percentage of macronutrient contributions [16,20], and lastly, the duration of supplementation differed between studies, with ZMA either administered chronically (i.e., 7 to 8 weeks and with measures pre- and post-intervention) or acutely, like the current investigation.

### 4.1. Limitations

The a priori sample size estimation predicted a sample of 36 male participants was required to assess the first hypothesis, that individuals who slept for 8 h per night would have better sleep quality and cognitive performance than those who slept for 4 h. Although thirty-six participants were recruited for each group, three withdrew out in the 4 h condition, although significant findings were observed for sleep opportunity; hence, sample size may not be a limitation. Actimetry may lack the sensitivity to detect changes in some sleep parameters such as sleep latency, due to the device being unable to distinguish the difference between movement of the wrist during sleep and general non-movement [57]. Polysomnography would offer a greater level of accuracy needed to detect meaningful change but comes with added time, expertise, time demand on the participant and cost. Further, participants conducted the experiments on the same days but weeks apart, so they had the same daily routine around work and study they had to attend during the day. However, we did not measure activity, which would have given information about daytime activities preceding sleep recordings across conditions. As we chose a young population of males, it limits the generalisability of the findings. Lastly, in the current investigation, females were omitted, further research should investigate the potential sex difference in supplementation effects after sleep loss where sleep (by polysomnography—to investigate maintenance of sleep) and the rhythms of mood, core body temperature tiredness and alertness (to investigate phase-advanced profiles in females vs. males) are measured.

### 4.2. Conclusions

The 8 vs. 4 h opportunity to sleep resulted in better “sleep” metrics as well as better Stroop scores. Our most important outcome was that ZMA did not improve markers of sleep quality or cognitive function after 2 nights of sleep loss when compared to a PLAC or a NoPill supplementation condition. Rather, in the ZMA condition, total sleep time was reduced by ~0.46 h compared to the other conditions. Further, an interaction was found where sleep time was reduced by ~0.94 h in the ZMA group in in the 8 h condition compared to NoPill or PLAC.

## 5. Practical Implications and Future Research

To the authors’ knowledge, this is the only study to investigate both habitual and partial sleep restriction on sleep and next-day cognitive performance relating to ZMA ingestion in large healthy male groups who either had 4 or 8 h sleep for 2 nights. The reduction in total sleep in the ZMA group may provide important recommendations and interventions for athletes who have high training/competition demands and are obtaining normal sleep and partial sleep deprivation. Further work should investigate the mechanisms of ZMA during sleep using polysomnography, considering a dose–response effect to consider changes in concentration of serum Zn/Mg status, which could alter findings specifically with sleep. With the several cognitive domains such as attention (division and switching of attention), perception and processing of information (fast reaction time) and visual–spatial skills (navigation in a virtual environment), these would be a subject for future research. Venous blood sampling would be integral to establishing both post-supplement mineral serum and ‘pre’ habitual Zn/Mg status. Differentiating habitually high Zn/Mg consumers from low consumers may offer an insight as to whether ZMA affects these groups to different extents. Lastly, a greater spread of levels of Zn and Mg supplementation including values below and above the required RDA would give a better understanding of the relationship of these essential minerals and resultant sleep.

## Figures and Tables

**Figure 1 behavsci-14-00622-f001:**
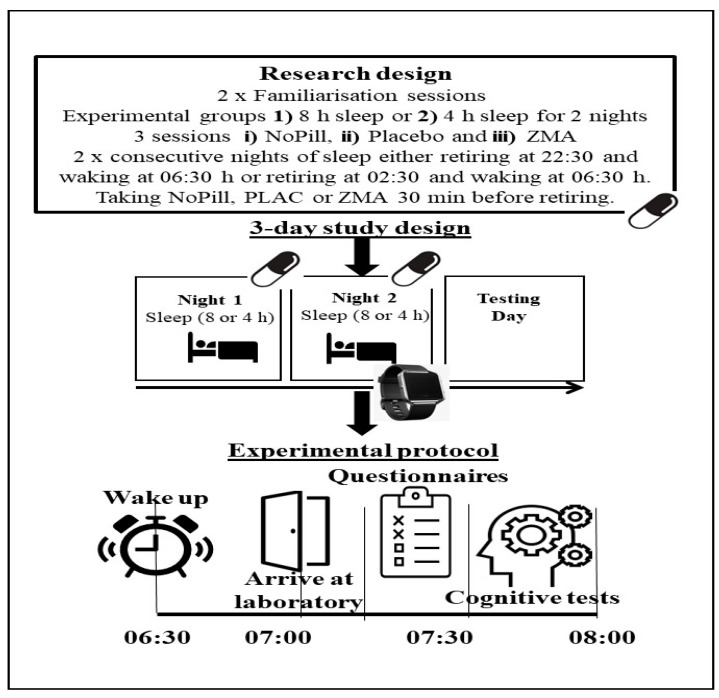
Schematic representation of the protocol undertaken by participants in the investigation. The watch represents Night 2 sleep recorded by actigraphy.

**Figure 2 behavsci-14-00622-f002:**
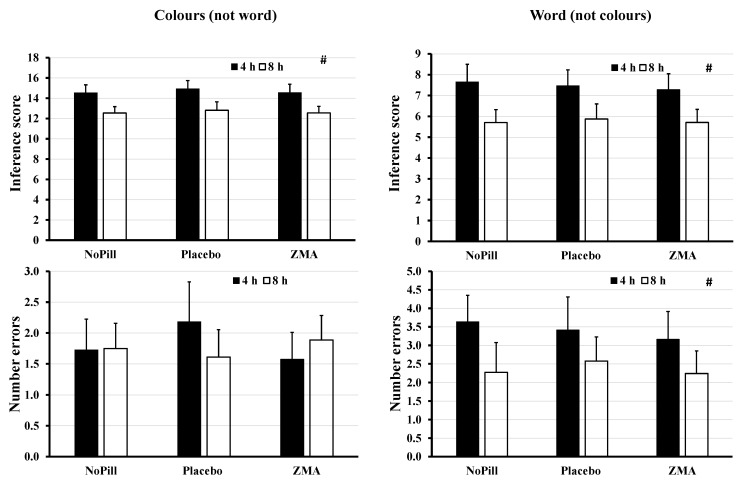
Mean (CI) for colours (not word) and word (not colours) for interference and errors for 8 vs. 4 h of sleep for the second night for each condition (NoPill, PLAC and ZMA). # represents a significant main effect for “sleep time”.

**Figure 3 behavsci-14-00622-f003:**
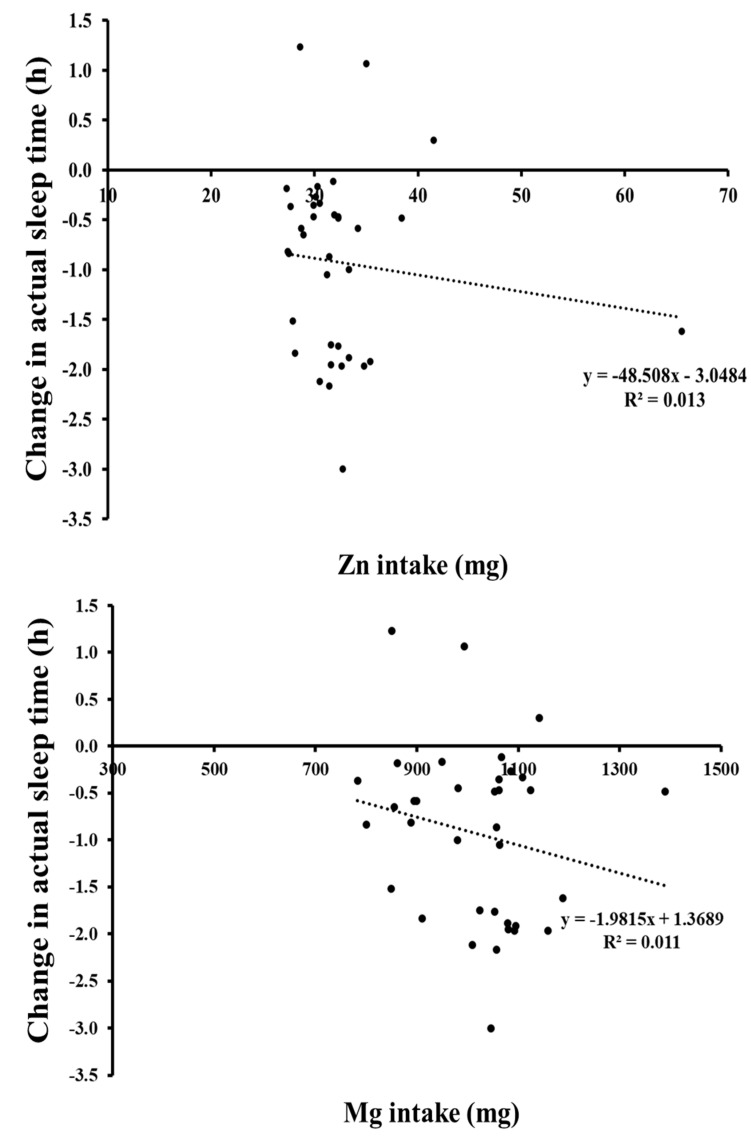
Relationship between change in total sleep time (decimal h) for ZMA-NoPill values at 8 h sleep, expressed per Zn and Mg intake (mg). The *x*-axis is set to the RDA amount.

**Table 1 behavsci-14-00622-t001:** Physical characteristics, baseline actimetry and baseline food intake of the participants, mean ± SD, as well as normal or recommended values from the literature. * Taken from NHS online reports [27]. # Taken from Boonstra et al. [28].

	4 h Sleep Group (n = 36)	8 h Sleep Group (n = 33)	Control and Recommended Values
**Physical Characteristics**	36 males	33 males	18 males and 21 females
Age (yr)	22.3 ± 1.8	23.1 ± 3.6	37.8 ± 9.5 #
Height (cm)	176.2 ± 6.0	176.3 ± 7.1	Not given #
Mass (kg)	78.9 ± 10.6	78.9 ± 9.4	Not given #
BMI (kg/m^2^)	25.4 ± 2.4	25.3 ± 2.9	18.5-24.9 *
**Baseline Actimetry**			**Control values**
Fragmentation (Au)	29.4 ± 13.3	29.2 ± 13.1	28.0 ± 1.0 #
Sleep efficiency (%)	82.0 ± 6.9	82.2 ± 7.1	86.0 ± 1.0 #
Actual sleep time (h:mm)	6:45 ± 00:45	6:42 ± 00:41	6:53 ± 00:06 #
Habitual retiring time (h:mm)	23:01 ± 00:32	23:17 ± 00:36	24:33 ± 00:07 #
Habitual wake time (h:mm)	7:09 ± 0:33	7:25 ± 0:36	8:24 ± 0:09 #
Sleep onset latency (h:mm)	0:12 ± 00:09	0:11 ± 00:10	0:10 ± 00:01 #
Time in bed (h:mm)	8:16 ± 00:31	8:08 ± 00:29	8:01 ± 0:07 #
**Baseline Food Intake**			**RDA**
Daily calories (kcal)	2544 ± 692	2507 ± 648	2500 *
Fats (g)	116 ± 78	117 ± 73	65 *
Protein (g)	163 ± 60	149 ± 56	56 *
Carbohydrates (g)	212 ± 93	215 ± 91	130 *
Zinc (mg)	13.7 ± 7.1	12.9 ± 6.4	11 *
Magnesium (mg)	450 ± 171	417 ± 124	400 *
Vitamin B6 (mg)	2.9 ± 1.2	3.0 ± 1.0	8 *

**Table 2 behavsci-14-00622-t002:** A comparison of the mean (±SD) values for actimetry and subjective sleep measures for two groups (8 vs. 4 h) for 3 conditions (NoPill, Placebo and ZMA) for the second night of sleep, with statistics given. Statistical significance (*p* < 0.05) is indicated in bold, and a trend (where 0.10 < *p* > 0.05) is indicate in italics.

Variable	Sleep 4 or 8 h	NoPill	PLAC	ZMA	Significance of between Effect for ‘Sleep Time’	Significance of Main Effect for ‘Pill’	Significance of Interaction
** *Actigraphy* **							
Sleep onset latency (decimal min)	4	8.6 ± 8.3	7.3 ± 8.0	8.4 ± 11.2	***F*_1.0, 67.0_, 7.085**, *p* = 0.010	*F*_1.7, 114.1,_ 0.731, *p* = 0.463	*F*_1.7, 114.1,_ 0.018, *p* = 0.971
	8	13.9 ± 13.3	12.6 ± 10.8	14.1 ± 9.3			
Sleep efficiency (%)	4	83.8 ± 6.5	84.6 ± 7.3	85.0 ± 7.0	*F*_1.0, 67.0_, 2.415, *p* = 0.125	*F*_1.7, 116.2,_ 0.639, *p* = 0.503	*F*_1.7, 116.2,_ 1.070, *p* = 0.337
	8	82.1 ± 7.3	82.5 ± 7.6	81.4 ± 7.7			
Actual sleep time (decimal h)	4	3.4 ± 0.2	3.5 ± 0.3	3.5 ± 0.3	***F*_1.0, 67.0_, 585.594**, *p* < 0.001	***F*_1.6, 108.9,_ 23.378, *p* < 0.001**	***F*_1.6, 108.9,_ 24.157, *p* < 0.001**
	8	6.5 ± 0.9	6.5 ± 1.0	5.6 ± 0.3			
Fragmentation index (Au)	4	22.9 ± 10.6	27.2 ± 11.8	26.5 ± 12.5	*F*_1.0, 67.0_, 2.783, *p* = 0.100	*F*_1.7, 113.4,_ 0.513, *p* = 0.570	*F_1.7, 112.6,_ 2.758, p = 0.079*
	8	30.5 ± 11.4	29.0 ± 10.8	28.3 ± 11.8			
** *Subjective Sleep Q’* **							
Ease to sleep?	4	0.5 ± 2.9	1.1 ± 2.7	0.3 ± 2.6	***F*_1.0, 67.0_, 6.197**, *p* = 0.015	*F*_1.3, 81.7,_ 0.155, *p* = 0.745	***F*_1.3, 81.7,_ 5.249, *p* = 0.019**
	8	−0.2 ± 2.0	−1.2 ± 2.4	−0.2 ± 2.0			
Time to sleep?	4	2.7 ± 2.4	2.6 ± 2.6	2.5 ± 2.8	***F*_1.0, 67.0_, 24.401**, *p* < 0.001	*F*_1.3, 91.6,_ 0.259, *p* = 0.679	*F*_1.3, 91.6,_ 0.161, *p* = 0.761
	8	0.1 ± 2.8	−0.1 ± 2.4	0.1 ± 2.8			
How well did you sleep?	4	0.6 ± 2.5	0.6 ± 2.4	0.9 ± 2.1	***F*_1.0, 67.0_, 5.765**, *p* = 0.019	*F*_1.4, 95.1,_ 0.114, *p* = 0.822	*F*_1.4, 95.1,_ 0.152, *p* = 0.784
	8	−0.4 ± 1.9	−0.3 ± 1.8	−0.4 ± 1.9			
What was your waking time?	4	−2.8 ± 1.8	−2.6 ± 1.8	−2.9 ± 1.8	***F*_1.0, 67.0_, 27.448**, *p* < 0.001	*F*_1.4, 94.8,_ 0.975, *p* = 0.353	***F*_1.4, 94.8,_ 3.777, *p* = 0.042**
	8	−0.8 ± 1.9	−0.8 ± 1.9	−0.5 ± 1.6			
Alertness 30-min after waking?	4	−1.8 ± 2.4	−1.7 ± 2.2	−2.0 ± 2.3	***F*_1.0, 67.0_, 18.642**, *p* < 0.001	*F*_2.0, 133.7_, 0.784, *p* = 0.454	*F*_2.0, 133.7_, 0.594, *p* = 0.546
	8	0.0 ± 1.4	−0.1 ± 1.8	−0.2 ± 1.5			
** *Stanford Sleep Q’* **							
Degree of sleepiness	4	4.2 ± 1.3	4.1 ± 1.3	4.1 ± 1.3	***F*_1.0, 67.0_, 40.736**, *p* < 0.001	*F*_2.0, 134.0_, 0.415, *p* = 0.661	*F*_2.0, 134.0_, 0.754, *p* = 0.473
	8	2.7 ± 0.7	2.8 ± 0.9	2.9 ± 0.9			

**Table 3 behavsci-14-00622-t003:** Mean (±SD) values for perceived onset of mood scores (POMS) and Stroop values for two groups (8 vs. 4 h) for 3 conditions (NoPill, Placebo and ZMA) for the second night of sleep, with statistics given. Statistical significance (*p* < 0.05) is indicated in bold, and a trend (where 0.10 < *p* > 0.05) is indicated in italics.

Variable	Sleep 4 or 8 h	NoPill	PLAC	ZMA	Significance of between Effect for ‘Sleep Time’	Significance of Main Effect for ‘Pill’	Significance of Interaction
** *POMS:* **							
Vigour	4	3.0 ± 2.6	3.3 ± 3.0	3.5 ± 3.2	***F*_1.0, 67.0,_ 13.096; *p* < 0.001**	*F*_1.8, 117.5_ 1.349; *p* = 0.262	*F*_1.8, 117.5_ 0.698; *p* = 0.482
	8	5.1 ± 3.1	5.9 ± 3.3	5.3 ± 2.7			
Anger	4	1.7 ± 1.7	1.9 ± 2.6	1.0 ± 2.2	*F_1.0, 67.0,_ 3.148; p = 0.081*	*F*_2.0, 134.0,_ 0.200; *p* = 0.819	** *F* ** **_2.0, 134.0,_ 4.114; *p* = 0.018**
	8	0.8 ± 1.7	0.6 ± 1.5	1.2 ± 2.3			
Tension	4	0.8 ± 1.3	0.6 ± 0.7	0.5 ± 0.9	***F*_1.0, 67.0,_ 6.805; *p* = 0.011**	*F*_1.7, 112.4,_ 1.832; *p* = 0.171	*F*_1.7, 112.4,_ 0.999; *p* = 0.359
	8	0.3 ± 0.8	0.4 ± 0.8	0.1 ± 0.2			
Calm	4	5.8 ± 3.1	6.7 ± 3.9	5.6 ± 3.9	*F*_1.0, 67.0,_ 1.087; *p* = 0.301	*F*_2.0, 134.0,_ 0.656; *p* = 0.521	*F* *_2.0, 134.0,_ 2.93; p = 0.057*
	8	7.3 ± 3.6	6.3 ± 3.2	6.6 ± 2.7			
Happy	4	4.1 ± 3.0	4.2 ± 3.3	4.0 ± 4.1	***F*_1.0, 67.0,_ 6.709; *p* = 0.012**	*F*_2.0, 133.7,_ 1.212; *p* = 0.301	*F*_2.0, 133.7,_ 0.530; *p* = 0.590
	8	5.6 ± 2.8	6.3 ± 2.9	5.3 ± 3.0			
Confusion	4	2.1 ± 2.9	2.0 ± 2.1	1.6 ± 1.5	***F*_1.0, 67.0,_ 8.891; *p* = 0.004**	*F*_1.8, 118.7,_ 0.604; *p* = 0.529	*F*_1.8, 118.7,_ 0.156; *p* = 0.831
	8	0.9 ± 2.0	0.9 ± 1.5	0.8 ± 1.6			
Depressed	4	1.7 ± 1.8	1.0 ± 1.4	1.5 ± 1.6	***F* _1.0, 67.0,_ 5.342; *p* = 0.024**	** *F* ** ** *_1.9 129.7,_ 4.272; p = 0.017* **	*F*_1.9, 129.7,_ 0.562; *p* = 0.566
	8	0.8 ± 1.7	0.4 ± 1.1	0.8 ± 1.6			
Fatigue	4	8.7 ± 3.7	8.2 ± 5.2	8.7 ± 3.6	***F*_1.0, 67.0,_ 30.384; *p* < 0.001**	*F* *_1.6, 108.1,_ 2.577; p = 0.092*	*F*_1.6, 108.1,_ 0.547; *p* = 0.543
	8	5.9 ± 1.8	4.4 ± 2.2	5.1 ± 2.5			
**Stroop**							
Colours number	4	57.2 ± 9.1	55.6 ± 9.8	57.5 ± 10.9	***F*_1.0, 67.0,_ 17.165; *p* < 0.001**	*F*_2.0, 134.0,_ 0.911; *p* = 0.405	*F*_2.0, 134.0,_ 0.342; *p* = 0.711
	8	68.0 ± 12.3	67.6 ± 15.7	68.0 ± 12.6			
Colour error	4	1.7 ± 1.5	2.2 ± 1.9	1.6 ± 1.3	*F*_1.0, 67.0,_ 0.085; *p* = 0.771	*F*_1.7, 111.1,_ 0.505; *p* = 0.570	*F_1.7, 111.1,_ 2.963; p = 0.065*
	8	1.8 ± 1.3	1.6 ± 1.4	1.9 ± 1.2			
Word number	4	133.1 ± 62.8	122.4 ± 57.2	137.7 ± 60.3	***F*_1.0, 67.0,_ 7.069; *p* = 0.010**	*F*_1.8, 117.7,_ 1.390; *p* = 0.253	*F*_1.8, 117.7,_ 0.940; *p* = 0.383
	8	184.8 ± 90.9	184.0 ± 93.0	184.9 ± 91.0			
Word error	4	3.6 ± 2.5	3.4 ± 2.0	3.2 ± 1.9	***F*_1.0, 67.0,_ 4.711; *p* = 0.034**	*F*_1.8, 122.5,_ 1.250; *p* = 0.288	*F*_1.0, 67.0,_ 0.997; *p* = 0.366
	8	2.3 ± 2.1	2.6 ± 2.6	2.2 ± 2.2			

## Data Availability

The data presented in this study are not available due to ethical restrictions.

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
