# Peer review of "In Males with Adequate Dietary Needs Who Present No Sleep Disturbances, Is an Acute Intake of Zinc Magnesium Aspartate, Following Either Two Consecutive Nights of 8 or 4 h of Sleep Deprivation, Beneficial for Sleep and Morning Stroop Interference Performance?"

_behavsci, 2024, doi:10.3390/bs14070622_

Round 1

Reviewer 1 Report

Comments and Suggestions for Authors

The present study aimed to examine the effects of acute zinc-magnesium aspartate (ZMA) supplementation on sleep, mood, and cognitive performance in young males under conditions of partial sleep deprivation or normal 8-hour sleep. ZMA supplementation showed no significant beneficial impact on the measured sleep parameters or cognitive performance metrics. This investigation largely replicates and extends the findings previously reported by Gallagher et al. (2024), with both studies reaching similar conclusions. The current results do not advance our understanding of ZMA supplementation's effects on sleep and performance beyond what was already known. Several methodological limitations warrant discussion:

  1. The study's exclusive focus on young males limits its generalizability. The omission of female subjects needs to be justified.  The conclusions need to be qualified to acknowledge the potential sex differences in the effects of ZMA supplementation..
  2. The authors should clarify whether this study and two related investigations (Gallagher et al., 2024 and 2023) were conducted independently, given their overlapping protocols and narrow time frame. Any potential data overlap between these reports should be explicitly addressed.
  3. As daytime activity can influence subsequent sleep, the authors should report whether actigraphy data showed any significant differences in participants' daytime activities preceding sleep recordings across conditions.
  4. The methodology for determining baseline food intake and instructions regarding diet during the experimental period require clarification. The authors should explain how zinc and magnesium intakes were assessed. Without objective measurements of daily zinc and magnesium intake, the correlations presented in Figure 3 lack a solid foundation. If such measurements were indeed performed, these findings should be moved from the Discussion to the Results section.
  5. Actimetry reference values: The "Normal or recommended values" presented in Table 1 should be contextualized. These appear to be control values from a single study and, while likely within normal ranges, should not be presented as recommended values (e.g., the 24:33 h retiring time).
  6. BMI of participants should be reported to provide a more complete picture of the study population.

Author Response

We thank the reviewer for their time and comments. We have allocated each question and answer a Q and A with number and given the page and lines numbers where the text has been changed.

Q1. The study's exclusive focus on young males limits its generalizability. The omission of female subjects needs to be justified.  The conclusions need to be qualified to acknowledge the potential sex differences in the effects of ZMA supplementation.

A1. We thank the reviewer for their comments. We agree and have amended the text to reflect this see below:

Page 1, line 2: Title: Now says males, abstract now has males in.

Page 2, line 82:  Aims. ... we examined in males,

Pages 3, lines 101--106: Materials and Methods. “Females were omitted from the current investigation to limit biological sex differences due to hormonal variation, as well as females being relatively phase advanced in their tiredness /alertness rhythms than that of males [22], with higher prevalence of difficulty maintaining sleep and early-morning awakenings reported in women vs. men. This could impact on morning mood states and cognitive performance especially after sleep loss, in a way different than males. As there is little research in this area, we sought to reduce bias.”

Page 13, lines 364-371: Limitations. “As we chose a young population of males it limits the generalizability of the findings. Lastly, in the current investigation females were omitted, further research should investigate the potential sex difference in supplementation effects after sleep loss where sleep (by polysomnography – to investigate maintenance of sleep) and the rhythms of mood, core body temperature tiredness and alertness (to investigate phase advanced profiles in female's vs men) are measured.”

Please note we intend to investigate SR and ZMA supplementation in females in the near future and morning Stroop and physical performance.

Q2. The authors should clarify whether this study and two related investigations (Gallagher et al. 2024 and 2023) were conducted independently, given their overlapping protocols and narrow time frame. Any potential data overlap between these reports should be explicitly addressed.

A2. We thank the reviewer for their comments The Gallagher et al. (2023) work concerns the intervention of naps rather and Gallagher et al. 2024 - ZMA supplementation. We have however, inserted in the method that 15 of the 69 participants data in the current investigation on ZMA and sleep has been previously published (Gallagher et al. 2024).

Page 2, lines 94-95. “Fifteen of the 69 participants data investigating ZMA and sleep has been previously published (Gallagher et al. 2024).”

Q3. As daytime activity can influence subsequent sleep, the authors should report whether actigraphy data showed any significant differences in participants' daytime activities preceding sleep recordings across conditions.

A3. We thank the reviewer for their comments. We agree, this is a good point. Unfortunately, we did not measure activity or actigraphy during the days of the experiments. What we can say however is that the participants conducted the experiments on the same days but weeks apart, so they had the same structure of lectures they had to attend during the day. They were also restricted in terms of levels of activity, food intake and were asked not to take caffeine. We have included this in the limitation section.

Page 13, lines 364-367: Limitations. “Further, participants conducted the experiments on the same days but weeks apart, so they had the same daily routine around work and study they had to attend during the day. However, we did not measure activity which would have given information about daytime activities preceding sleep recordings across conditions.”

Q4. The methodology for determining baseline food intake and instructions regarding diet during the experimental period require clarification. The authors should explain how zinc and magnesium intakes were assessed. Without objective measurements of daily zinc and magnesium intake, the correlations presented in Figure 3 lack a solid foundation. If such measurements were indeed performed, these findings should be moved from the Discussion to the Results section.

A4. We thank the reviewer for their comments. We have clarified the methods for determining baseline food intake and instructions regarding diet during the experimental period. We apologise for missing this out, we have inserted this information in the text and the reference in the reference section– see below.

Page 4, lines 128-130: Research design.  “The values for measurements of daily zinc and magnesium were determined by dietary analysis using the computer program Nutritics (Nutritics, Ireland [29]), this process was conducted by a SENr registered Sports and Exercise Nutritionist.”

For the second part of your question. We have put a section in the results, moved Figure 3 and changed the discussion see below.

Page 6, lines 237-241: Results.

3.5. Pearson correlations between sleep difference (h) and Mg or Zn levels (either mg or mg/kg body mass) for ZMA-NoPill for 8 h Sleep condition.

A small negative correlation (r = -0.203, p=0.236) which was not significant was found for Mg and a trivial negative correlation (r = -0.106, p=0.538), again not significant was found for Zn. The strength of the relationship was not affected when controlling for body mass for Mg (r = -0.213, p=0.220) or ZN (r = -0.178, p=0.306).

Page 12, lines 317-324. Discussion. No significant correlations were found for Mg or Zn (r = -0.203, p=0.236; r = -0.106, p=0.538) and change in total sleep between ZMA and NoPill values for the 8-h condition (that showed the >sleep loss with ZMA ~0.94 h; Figure 3). Zn and Mg levels ingested through diet and supplementation were ~31.9 mg or 336 % RDA for Zn and ~1016.6 mg or 339 % RDA for Mg. The strength of the relationship was not affected when controlling for body mass for Mg (r = -0.213, p=0.220) or Zn (r = -0.178, p=0.306). Where 1.3 and 1.1 % of the variation in Zn or Mg intake can account for change in actuals sleep time (Figure 3). In future research, greater spread of levels of Zn and Mg supplementation including values below and above the required RDA would give a better understanding of the relationship of these essential minerals and resultant sleep.

Q5. Actimetry reference values: The "Normal or recommended values" presented in Table 1 should be contextualized. These appear to be control values from a single study and, while likely within normal ranges, should not be presented as recommended values (e.g., the 24:33 h retiring time).

BMI of participants should be reported to provide a more complete picture of the study population.

A5. We thank the reviewer for their comments and have reported them as control values and recommended values. Page 3. Whilst we have given information on height and weight so BMI could be calculated we have also reported this in the Table 1, Page 3. We do not generally report BMI values as the kg/body weight does not reflect fat or fat-free mass and can lead to confusion where muscular trained individuals are classed as being obese and overweight.

Reviewer 2 Report

Comments and Suggestions for Authors

The introduction is centered on sleep deprivation and supplementation with Zinc-Magnesium-Aspartate (ZMA).  However, there is little theoretical support for the use of the Stroop test and how it is used to identify cognitive performance.

How do they conceptually and operationally define cognitive performance?  Although the Stroop test has been widely used to assess cognitive functions such as attention, processing speed, cognitive flexibility and working memory among others, it is important that you define which processes you are considering in your assessment.

In the method section, it is important to define the purpose of the test application and the type of analysis to be performed, since interference results are not presented, only completed words and errors.

The Stroop Test is widely used to measure the ability to inhibit cognitive interference.  Although the method is described as Stroop (word color interference test), the interference analysis as such is not performed.

There are several methods and formulas to calculate the interference, so I suggest to make an analysis of this parameter.

The study is relevant with respect to sleep deprivation and supplementation; however, it needs to give more relevance to the cognitive performance section.

It is also suggested to make correlations with the Stroop test interference, because it is concluded that supplementation improves performance in this test.

As well as discussing in terms of what they are going to conceptualize as cognitive performance when using the Stroop test.

Author Response

We thank the reviewer for their time and comments. We have allocated each question and answer a Q and A with number and given the page and lines numbers where the text has been changed.

Q1. The introduction is centered on sleep deprivation and supplementation with Zinc-Magnesium-Aspartate (ZMA). However, there is little theoretical support for the use of the Stroop test and how it is used to identify cognitive performance.

A1. We thank the reviewer for their comments, we agree and have corrected the text to reflect this.

Page 1, line 5: Title. ….beneficial for sleep and morning Stroop-interference performance.

Page 2, lines 54-60: Introduction. “This can have detrimental effects on mood and motivation, an essential element for tasks requiring higher cognitive function (such as executive functions) especially in the morning when this is compromised by partial sleep deprivation [8,11]. Executive functions notably include the ability to plan and coordinate considered action regardless of alternatives, to monitor action, update as necessary and to suppress distractions by focusing attention on the relevant information (i.e., inhibition). One of the most widely used neuropsychological tests to study attention and notably its inhibitory processes is the Stroop Colour-Word test [12,13]. This task has been used extensively to study limitations in the ability to fully suppress the influence of a dominant source of information, such as automatic word reading. Therefore, the magnitude of the Stroop interference has been used as an indicator of the efficiency of the inhibitory function.”

Page 11, lines 280-286: “Although the pre-frontal cortex may be susceptible to the effects of sleep loss, research investigating sleep deprivation on executive functional tasks (measures of prefrontal functioning) show inconsistent findings. Sleep loss may affect specific cognitive systems above and beyond the effects produced by cognitive declines or impaired attentional processes [42]. The two attentional systems involved in the Stroop task are i) the anterior attentional system associated with the dorsolateral prefrontal cortex – assists naming/task relevant information and other executive functions and ii) the Posterior attentional system (anterior cingulate cortex) – selects appropriate responses and allocates attentional resources.”

Q2. How do they conceptually and operationally define cognitive performance?  Although the Stroop test has been widely used to assess cognitive functions such as attention, processing speed, cognitive flexibility and working memory among others, it is important that you define which processes you are considering in your assessment.

A2. We thank the reviewer for their comments. We have amended the text in the introduction and aims to reflect this to consider the task as one investigative executive function, specifically attention and notably its inhibitory processes - an indicator of the efficiency of the inhibitory function. We have changed the following sections to reflect this.

Page 1, lines 4-5: Title. beneficial for sleep and morning Stroop-interference performance.

Page 2, lines 52-60: Introduction. “This can have detrimental effects on mood and motivation, an essential element for tasks requiring higher cognitive function (such as executive functions) especially in the morning when this is compromised by partial sleep deprivation [8,11]. Executive functions notably include the ability to plan and coordinate considered action regardless of alternatives, to monitor action, update as necessary and to suppress distractions by focusing attention on the relevant information (i.e., inhibition). One of the most widely used neuropsychological tests to study attention and notably its inhibitory processes is the Stroop Colour-Word test [12,13]. This task has been used extensively to study limitations in the ability to fully suppress the influence of a dominant source of information, such as automatic word reading. Therefore, the magnitude of the Stroop interference has been used as an indicator of the efficiency of the inhibitory function.”

Page 2, lines 86-88. Aims. .... b) cognitive performance - attention and notably its inhibitory processes ([19] Stroop task/test) in our chosen population of healthy male recreational athletes irrespective of the prior sleep taken for 2 nights.

Q3. In the method section, it is important to define the purpose of the test application and the type of analysis to be performed, since interference results are not presented, only completed words and errors.

A3. We thank the reviewer for their comments we have now inserted information regarding the interference tests in the introduction and aims – see answer 2. We have also added information of calculation of Interference in the measurement section with the equation.

Page 4, lines 165-171: Measurement. “The raw data were analysed for number of errors (to give an indication of pacing/ speed accuracy in the 45 s) interference scores (an indicator of the efficiency of the inhibitory function), where C represents the correct answers produced in 45 s in naming colours and CW corresponds to the correct answers achieved in 45 s in the interference condition [34]. This was also done for Naming of Word not Colour test.

I = [(C − CW)/(C + CW)) × 100] ……Equation 1”

Q4. The Stroop Test is widely used to measure the ability to inhibit cognitive interference.  Although the method is described as Stroop (word color interference test), the interference analysis as such is not performed.

A4. We thank the reviewer for their comments. We apologies for not initially being remiss in conducting this analysis, we fully agree with your highlighting missing components of the Stroop test. Originally, for the word/colour interference test) we measured 1) Naming word test, 2) Naming colour test and 3) Naming colour of word test and 4) Naming of colour test (see text below from methods), this was filmed and the total and errors written in a logbook. This would have enabled us to analysis interference according to recommended means – such that (i) the number of correct answers in a fixed time in each SCWT condition (W, C, CW) and (ii) a global index relative to the CW performance minus reading and/or colours naming abilities are computed (Scarpina F and Tagini S (2017) The Stroop Colour and Word Test Front. Psychol. 8:557. doi: 10.3389/fpsyg.2017.00557). Unfortunately, the current experiment was abruptly interrupted by COVID restrictions (we were given 20-min to leave the building) and when we were allowed back into the laboratory hard copies of the Naming word test along with other files had been binned as the university had converted the lab into a PPE and COVID testing swab storage area. We continues testing after this date and have given the updated ethics date and number which included the COVID-19 governmental guidelines.

We do have the Naming colour test data and so we have used the formula of Valgimigli et al. (2010; The Stroop test: a normative Italian study on a paper version for clinical use. G. Ital. Psicol. 37, 945–956. doi: 10.1421/33435), Where DC represents the correct answers produced in 45 s in naming colours and DI corresponds to the correct answers achieved in 45 s in the interference condition.

I = ((DC − DI)/(DC + DI)) × 100

We have re-run the analysis and included the inference scores in a new Figure 2. And included the new results on page 5 and integrated these in the discussion.

We have also amended the abstract and discussion to represent this.

Page 1, lines 23/24: Abstract. ….lower colour-interference and word-interference scores.

Page 11, lines 261-263: Discussion….. Further, morning Stroop Interference scores an indicator of the efficiency of the inhibitory function were better for the 8-h condition, colour-word and word-colour values being lower and word error lower.

These findings were expected in agreement with previous literature investigating the effect of partial sleep deprivation by restriction on sleep, as well as some measures of verbally mediated processing speed and executive functioning [8,40].

Q5. There are several methods and formulas to calculate the interference, so I suggest to make an analysis of this parameter.

A5. We thank the reviewer for their comments – we agree. We hope we have now answered this in Answer 4.

Q6. The study is relevant with respect to sleep deprivation and supplementation; however, it needs to give more relevance to the cognitive performance section.

A6. We thank the reviewer for their comments. We have added a section to the introduction, and have added the new results for interference to the discussion with an update of the findings in reference to cognitive performance - attention and notably its inhibitory processes.

Page 11, lines 261-266: Discussion. “Further, morning Stroop Interference scores an indicator of the efficiency of the inhibitory function were better for the 8-h condition, colour-word and word-colour values being lower and word error lower. These findings were expected in agreement with previous literature investigating the effect of partial sleep deprivation by restriction on sleep, as well as some measures of verbally mediated processing speed and executive functioning [8,40].”

Page 11, lines 280-295: Discussion.

“In agreement with the current findings the Stroop task has been shown to be sensitive to 100% sleep loss (after a total of 24 and 36 h total sleep deprivation) in male or young healthy males and female participants [43,44]. Further, in the current investigation we found morning Colour-word Stroop Interference scores (indicator of the efficiency of the inhibitory function) were better for the 8-h than 4-h condition. But we also showed the reverse Stroop (Word-colour) Interference values to be sensitive to sleep loss, where the participants read the word that was printed in a different colour to the word. These levels of inference were lower than the Colour word test as expected as the bases of the “Stroop effect” relies on humans have trouble naming a physical colour when it is used to spell the name of a different colour – such that there is a delay in reaction time between congruent and incongruent stimuli. Lastly, we found with the 8 h sleep not only was Word-colour interference values lower but also the error, hence an associated accuracy and speed reduction with sleep loss. On the other hand, others have found no effect of short-term sleep deprivation on the Stroop task after 34-36 h total sleep deprivation, suggesting that sleep deprivation does not selectively impair prefrontal functioning, notably the cognitive flexibility and the capacity to shift from one response set to another [45,46]. However, in the supporting literature confounding factors such as fatigue, stress, and different amounts of sleep debt as well as the method used to analysis Inference or type of Stroop (Colour-Word, Emotional and Specific) as well as using a fixed time (or not) might have affected performance.”

Q7. It is also suggested to make correlations with the Stroop test interference, because it is concluded that supplementation improves performance in this test.

As well as discussing in terms of what they are going to conceptualize as cognitive performance when using the Stroop test.

A7. We thank the reviewer for their comments. Part B to answer your questions - We have considered cognitive performance regarding the Stroop test to be regarding verbally mediated processing speed and executive functioning (new section in introduction and methods) specifically relating to attention and notably its inhibitory processes.

Answer to part A of your question - We did not find any main effect, or interactions for Stroop and ZMA group either in the results section or the conclusion. We did find ZMA deducted sleep time though (negative effects). As we found no effect, we do not see the need for correlations.

In the results 3.2.1 lines 201-208. We say "There was a significant group effect for opportunity to sleep (“sleep time”) where sleep onset latency (mean difference of 5.4 decimal min; 95% CI: 1.3–9.4, p<0.010, d=0.55), actual sleep time (2.74 decimal h; 95% CI: 2.5–3.0, p<0.001, d=1.79) were higher at 8 h than 4 h opportunity (Table 2). There was a significant main effect of “Pill” for actual sleep time where ZMA condition led to a reduced sleep time compared to NoPill (0.44 decimal h; 95% CI: 0.2–0.7, p<0.001, d=0.26) and a reduced sleep time compared to PLAC (0.49 decimal h; 95% CI: 0.3–0.7, p<0.001, d=0.30). There was a significant interaction of “Sleep” and “Pill” for actual sleep time where profiles of 4 and 8 h for NoPill and PLAC are similar but during the 8 h opportunity to sleep is reduced in the ZMA condition compared to NoPill and PLAC (~0.94 h, d=3.08, Table 2).”

And in the conclusion lines 373-377. “The 8 vs 4-h opportunity to sleep resulted in better “sleep” metrics as well as better Stroop scores. Our most important outcome was that ZMA did not improve markers of sleep quality, cognitive function after 2-nights of sleep loss when compared to a PLAC or a NoPill supplementation condition. Rather in the ZMA condition total sleep time was reduced by ~0.46 h compared to the other conditions. Further, an interaction was found where sleep time was reduced by ~0.94 h in the ZMA group in in the 8 h condition compared to NoPill or PLAC."

Round 2

Reviewer 2 Report

Comments and Suggestions for Authors

The authors addressed all the comments made on the original document.